# Future Perspectives in the Second Line Therapeutic Setting for Non-Oncogene Addicted Non-Small-Cell Lung Cancer

**DOI:** 10.3390/cancers15235505

**Published:** 2023-11-21

**Authors:** Marco Siringo, Javier Baena, Helena Bote de Cabo, Javier Torres-Jiménez, María Zurera, Jon Zugazagoitia, Luis Paz-Ares

**Affiliations:** 1Department of Medical Oncology, 12 de Octubre Hospital, 28041 Madrid, Spain; marco.siringo@uniroma1.it (M.S.); javier.baena@salud.madrid.org (J.B.); helenabdc@usal.es (H.B.d.C.); javier.torres@salud.madrid.org (J.T.-J.); mzureberj@gmail.com (M.Z.); 2Department of Medical Oncology, Sapienza University of Rome, 00100 Rome, Italy; 3Lung Cancer Clinical Research Group, Spanish National Cancer Research Center (CNIO), 28029 Madrid, Spain; 4Ciberonc, 28029 Madrid, Spain; 5Medicine Department, Medicine Faculty, Complutense University, 28040 Madrid, Spain

**Keywords:** second line NSCLC, antibody–drug conjugates, immunotherapy combinations

## Abstract

**Simple Summary:**

Despite the use of novel agents in the first-line therapeutic setting, such as PD-1/PDL1 axis blockers for non-oncogene addicted non-small-cell lung cancer, most patients with advanced disease experience progression will succumb to the illness within a short period of time. Currently, the standard second-line treatment consists primarily of systemic cytotoxic therapies, which typically yield poor outcomes. Recently, several novel therapeutic strategies have emerged that may improve patient outcomes. This article reviews current state-of-the-art treatments in this scenario and highlights potential future options.

**Abstract:**

Immune checkpoint inhibitors (ICIs) have revolutionized the management of non-oncogene addicted non-small-cell lung cancer (NSCLC). Blocking the anti-PD-1 axis represents the current standard of care in the first-line setting, with drugs administered either as monotherapy or in combination with chemotherapy. Despite notable successes achieved with ICIs, most of their long-term benefits are restricted to approximately 20% of patients. Consequently, the post-failure treatment landscape after failure to first-line treatment remains a complex challenge. Currently, docetaxel remains the preferred option, although its benefits remain modest as most patients do not respond or progress promptly. In recent times, novel agents and treatment combinations have emerged, offering fresh opportunities to improve patient outcomes. ICIs combined either with antiangiogenic or other novel immunotherapeutic compounds have shown promising preliminary activity. However, more mature data concerning specific combinations do not support their benefit over standard of care. In addition, antibody–drug conjugates seem to be the most promising alternative among all available compounds according to already-published phase I/II data that will be confirmed in soon-to-be-published phase III trial data. In this report, we provide a comprehensive overview of the current second-line treatment options and discuss future therapeutic perspectives.

## 1. Introduction

Lung cancer is a leading cause of cancer-death worldwide, with non-small-cell lung cancer (NSCLC) accounting for 80–85% of cases. More than half of patients are diagnosed with advanced disease, which even nowadays results in an incurable condition for most patients [1,2].

Increasing molecular and tumor biology knowledge has notably changed cancer treatment paradigms. Notably, the discovery of genomic aberrations dictating oncogene addiction—such as *EGFR* mutations, *ALK* rearrangements, and others—has led to a major revolution not only in the prognosis but also in the quality of life of this subgroup of patients. Regrettably, scarcely 40–60% of NSCLC have driver alterations amenable for targeted therapeutic interventions [3].

Immune-checkpoint inhibitors (ICIs), particularly programmed cell death-1 (PD-1) axis blockers, have revolutionized the management of many solid tumors. For non-oncogene addicted NSCLC, PD-1 axis blockade represents the current standard of care in the first-line setting, either as monotherapy or in combination, depending largely on programmed cell death ligand-1 (PD-L1) expression in tumor cells (TPS) [4].

Pembrolizumab, a humanized monoclonal anti-PD-1 antibody, demonstrated consistent overall survival (OS) benefit [HR:0.62 (0.48–0.81), median OS 26.3 versus 13.4 months], compared to platinum doublet chemotherapy in patients with PD-L1 expression in more than 50% of tumor cells (PD-L1 > 50%). These data were subsequently confirmed with longer follow-up and validated in other trials with alternative PD-1/PD-L1 axis blockers such as atezolizumab (anti-PD-L1) and cemiplimab (anti PD-1), with similar outcomes [5,6,7].

Chemotherapy nevertheless remains an important component of the treatment strategy, particularly in patients with low PD-L1 expression (<50%). Through its immunogenic properties, platinum-based chemotherapy can empower the effect of immunotherapy, inducing immunogenic death and modulating the immunosuppressive landscape of the tumor microenvironment (TME) [8]. For these reasons, ICIs have been successfully developed in combination with platinum-based chemotherapy as a first-line treatment, with almost half of the treated patients alive after two years. Platinum doublets with pemetrexed or paclitaxel, according to histology, plus ICIs represents the standard of care in patients without tumor PD-L1 overexpression. The KEYNOTE 189 and 407 trials demonstrated efficacy of chemotherapy associated with pembrolizumab, while the Checkmate 9LA trial evaluated nivolumab and ipilimumab with a short course of chemotherapy, for 2 cycles, with a favorable risk-benefit profile [9,10,11,12].

Despite the relevant results with first-line ICIs in advanced NSCLC, primary and secondary resistance still occurs, while, unfortunately, only a small proportion of patients demonstrate sustained benefit from immunotherapy. This translates into a 5-year OS rate that ranges between 31% and 12.5% according to PD-L1 status and histology. The corresponding figures for progression-free survival (PFS) are 7.5–10.8%, which underscores the need to fully understand the mechanisms of primary and adaptive resistance and to develop effective salvage therapies [10,11,12].

Nowadays, the standard of care second-line treatment for NSCLC patients consists mainly of cytotoxic agents either as monotherapy or in combination with other drugs. This review aims to provide the state-of-the-art treatment approaches in this clinical context and to highlight potential options after progression on first-line treatments, including ICIs, for patients with non-oncogene addicted metastatic NSCLC.

## 2. Materials and Methods

This is a non-systematic review of current concepts in second-line treatments for non-oncogene addicted metastatic NSCLC. References were identified through searches of PubMed using the terms “second-line NSCLC” (4266 results; 624 clinical trials), “NSCLC immunotherapy resistance” (848 results; 27 clinical trials), and in clinical trial registries (clinicaltrials.gov). Articles were selected based mainly on their clinical applicability, with priority given to practice-changing clinical trials or relevant translational or comprehensive reviews published in the last five years. Relevant articles were also identified through searches of the authors’ files and bibliographies of papers. Unpublished reports from scientific conferences (ASCO, ESMO, WLCC, STIC) were identified from abstract books. Only articles published in English were included. All the references cited in this article were reviewed.

## 3. Results

### 3.1. Chemotherapy ± Antiangiogenic Agents

#### 3.1.1. Chemotherapy

Second-line chemotherapy is generally proposed following a combination of first-line chemotherapy plus immunotherapy, administered either concurrently or sequentially. Inferring data from the pre-ICI era, docetaxel is considered the standard of care, while alternative solutions could be represented by nab-paclitaxel, gemcitabine and vinorelbine [13,14,15,16,17]. In addition, in specific circumstances or within certain geographically defined regions, pemetrexed or tegafur/gimeracil/oteracil (S-1) may, respectively, emerge as alternative options as well [18,19].

Interestingly, considerable evidence suggests that prior immunotherapy could enhance the efficacy of subsequent cytotoxic agents, improving tumor control over time. As an example, in a study of 77 patients, the overall response rate (ORR) to chemotherapy administered after immunotherapy was significantly higher compared to the same chemotherapy regimen administered in patients who did not receive prior immunotherapy (66.7–39.5%) [20]. Another study also demonstrated that second-line chemotherapy outcomes were better for patients who achieved disease control with prior immunotherapy (ORR 66.7% versus 16.7%) [21]. Notably, outcomes with docetaxel were improved in recent clinical trials compared to those carried out prior to the introduction of ICIs, as shown in the comparator arm of the Codebreak-200 study (ORR: 13.2%; median PFS: 4.5 months; median OS: 11.3 months) [22].

This favorable “post-immunotherapy-effect” could be explained by the influence of chemotherapy, which can modulate immune cells, thereby increasing antitumor responses while inhibiting immunosuppressive cells [23]. Importantly, the half-life of ICI monoclonal antibodies (mAbs) is in the range of 3 weeks, meaning that relevant concentrations still circulate for some 100 days following the last administration, potentially inducing a transient synergism with the new chemotherapy line [24].

#### 3.1.2. Chemotherapy Plus Anti-Angiogenic Agents

Antiangiogenic agents and other multi-tyrosine-kinase-inhibitors (TKIs) have been tested in the second-line scenario, typically in combination with chemotherapy before the availability of ICIs, although some studies have also been performed in the post-ICI era. 

Nintedanib is a triple-angiokinase inhibitor that occupies adenosine triphosphate (ATP) binding sites in the kinase domain of pro-angiogenic receptors (VEGFR-1, VEGFR-2, VEGFR-3, FGFR-1, FGFR-2, FGFR-3, PDGFR-α, PDGFR-β). This compound inhibits downstream signaling pathways, inducing apoptosis and proliferation arrest in cells associated with angiogenesis [25,26]. Moreover, nintedanib could inhibit immunosuppressive cells such as cancer-associated fibroblasts and facilitate the accumulation and activation of intertumoral CD8+ T-cells [27]. 

In the phase III LUME-Lung-1 trial, the docetaxel/nintedanib combination was compared to docetaxel/placebo as second-line treatment for NSCLC. Regardless of histology, PFS was significantly improved in the nintedanib arm, but OS was similar in the overall population [HR: 0.94 (0.83–1.05); median OS 10.1 versus 9.1 months]. However, OS was statistically significantly improved in the nintedanib group [HR:0.83 (0.70–0.99); median OS 12.6 versus 10.3 months] for those patients with adenocarcinoma and in patients with short PFS during first-line treatment [HR: 0.75 (0.60–0.92); median OS 10.9 versus 7.9 months] [28]. This combination was also analysed in a non-interventional study of 137 patients undergoing second-line treatment of adenocarcinoma in a post-chemotherapy-immunotherapy failure scenario. The results of an interim analysis confirmed a disease control rate (DCR) of 72.5% and a median PFS of 4.8 months with an acceptable safety profile. These benefits were numerically higher in patients who were treated first-line for less than 9 months. However, these data are preliminary, and a mature OS analysis is awaited [29].

Ramucirumab, a human IgG1 mAb that targets the extracellular domain of VEGFR-2, has also been tested. In the phase III Revel trial, the addition of ramucirumab to docetaxel improved outcomes compared with docetaxel alone. However, the median OS increase was just 1.4 months [HR:0.86 (0.75–0.98)] [30]. 

Weekly paclitaxel plus bevacizumab was compared to docetaxel in the Ultimate phase III trial, demonstrating better median PFS [HR:0.61 (0.44–0.86); 5.4 versus 3.9 months] and ORR (22.5% versus 5.5%). Despite the combination showing a similar median OS (HR:1.17, *p* = 0.50), paclitaxel plus bevacizumab exhibited a lower incidence of grade 3/4 toxicities (45.9% versus 54.5%) including neutropenia (19.3% versus 45.4%). Of note, grade 3/4 neuropathy (8.3% versus 0%) and hypertension (7.3% versus 0%) were more common in the experimental arm [31] (Table 1).

These data are in line with a retrospective study, where 67 unselected NSCLC patients were treated with docetaxel plus ramucirumab after the failure of second-line immunotherapy. The observed ORR was 36%, with a DCR of 69%, a median PFS of 6.8 months, and a median OS of 11.0 months. These findings suggested that anti-angiogenic agents after ICIs as third-line therapy seemed to be superior compared to second-line outcomes, supporting a synergistic interaction of ICIs and anti-angiogenic agents [32]. A systematic review of 10 studies suggests a numerical better PFS of docetaxel plus ramucirumab in prior exposed patients to ICI than in those naïve to these compounds (5.7 vs. 3.8 months) with statistical significance in two trials (*p* = 0.012 and *p* = 0.041). However, according to this publication, docetaxel plus ramucirumab could not offer OS advantage in ICI pretreated patients [33]. Hence, another retrospective observational study, including 1439 patients, reported that patients treated with chemotherapy with or without anti-angiogenics after ICIs had better ORR [HR 1.71 (1.19–2.46); *p* = 0.004] without any OS [HR 1.05 (0.86–1.28), *p* = 0.63] and PFS advantage [HR 0.95 (0.8–1.12), *p* = 0.55], compared to patients ICI-untreated [34].

Overall, the Anselma meta-analysis of 8629 individual patients revealed that the benefit of anti-angiogenic agents appears to be independent of the type of compound, with modest OS [HR: 0.93 (0.89–0.98), *p* = 0.005] and PFS [HR: 0.80 (0.77–0.84), *p* < 0.0001] outcomes obtained. However, the benefit may be increased in younger patients and in those with refractory tumors. This study also manifested a good and manageable safety profile [35].

Several studies have focused on the relationship between angiogenesis and immunosuppression, suggesting that anti-angiogenics could be relevant for overcoming immunoresistance [36,37]. One proposed mechanism is that anti-angiogenic agents might enhance the “angio-immunogenic switch”. This term refers to an increase in tumor infiltration by immune cells in the TME. Tumor-blood vessel reduction favors perfusion and oxygenation, and as a result a better penetration of chemotherapy and ICI drugs into the TME, which results in a better penetration of immune cells [38,39].

#### 3.1.3. Antibody-Drug Conjugates (ADCs)

Antibody-drug conjugates (ADCs) combine the specificity of mAbs with the cytotoxic effects of chemotherapy. ADCs consist of an antibody connected to a payload via a linker. Antibodies usually target cell-surface proteins that undergo rapid internalization and mediate intracellular unloading of the cytotoxic payload; IgG1 is the preferred option as this antibody has the strongest antibody-dependent cellular cytotoxicity (ADCC) [40]. Payloads are most typically chemotherapeutic compounds, potent DNA-damaging agents, or tubulin polymerization inhibitors. Finally, the type of linker may contribute to a “bystander effect”. This mechanism favors the killing effect of the payload on neighboring cancer cells that do not express the target antigen. ADCs must be stable enough to circulate to the cancer tissue and specific enough to bind only to tumor-associated antigens [41].

### 3.2. Anti-TROP2 ADCs

Trophoblast cell surface antigen-2 (Trop-2) is a transmembrane calcium signal transducer glycoprotein that is highly expressed in various human epithelial malignancies. Trop-2 expression has been correlated with progression and development of metastasis [42].

Datopotamab-Deruxtecan (Dato-DXd) is an ADC comprising a humanized anti-TROP2 IgG1 antibody linked to a topoisomerase-I inhibitor via a stable tetrapeptide-based cleavable linker. The Phase I Tropion pan-tumor-01 demonstrated a durable antitumor activity and a manageable safety profile in NSCLC patients pretreated with Dato-Dxd. Two hundred and ten patients, with a median of three prior treatment lines, were included. Responses to Dato-Dxd occurred regardless of TROP-2 expression, with an ORR of 26%, median duration of response (DoR) of 10.5 months, median PFS of 6.9 months, and median OS of 11.4 months. Interestingly, the immunohistochemical expression of this glycoprotein on tumor cells is not considered a predictive biomarker for response. Grade ≥3 treatment-emergent adverse events (AEs) and treatment-related adverse events (TRAEs) occurred in 54% and 26% of patients, respectively, with interstitial lung disease occurring in 6% of patients [43]. The phase III Tropion LUNG-01 trial, which compared Dato-DXd to docetaxel in patients previously treated with ICIs, showed positive results with an ORR of 26.4% vs. 12.8%, a PFS of 4.4 vs. 3.7 months [HR: 0.75 (0.62–0.91); *p* = 0.004] with a trend of advantage in OS [44].

Sacituzumab-Govitecan is an ADC composed of an anti-Trop-2 coupled to the cytotoxic SN-38 payload, an active metabolite of irinotecan. The antibody and the payload are joined via a proprietary, hydrolysable linker [45]. The compound was evaluated in 54 heavily pre-treated (including ICIs) patients with metastatic NSCLC within the phase I/II IMMU-132-01 trial. Sacituzumab-govitecan demonstrated moderate activity with an ORR of 19%, an mDoR of 6.0 months and a clinical benefit rate of 43%. The median PFS and median OS in the intention-to-treat-population were 5.2 months and 9.5 months, respectively, with a manageable safety profile (most common grade 3 AE was neutropenia in 28% of patients). Again, TROP-2 immunochemistry expression did not confirm predictivity of clinical benefit. The phase III Evoke trial is currently underway to evaluate the efficacy and safety of sacituzumab-govitecan versus docetaxel in NSCLC patients who progressed on chemotherapy and ICIs [46].

### 3.3. Anti-CEACAM5 ADCs

Carcinoembryonic-antigen-related cell adhesion molecule 5 (CEACAM5), a cell-surface glycoprotein, is overexpressed in non-squamous-NSCLC, where approximately 20–30% of patients exhibit moderate/strong CEACAM5 levels detected by immunochemistry [47].

Tusamitamab-ravtansine is an ADC that selectively targets CEACAM5-expressing tumor cells. The mAb is covalently linked to a potent cytotoxic maytansinoid (DM4). Ninety-two pretreated NSCLC patients were exposed to tusamitamab-ravtansine in the NCT02187848 study. This trial was designed with two cohorts based on CEACAM5 expression levels by immunochemistry ≥2+ in moderate expressors (intensity between ≥1% to <50% of tumor cell population) and high expressors (≥50% of the tumor cell population). Despite only 7.1% of patients showing a response in the moderate-expression cohort, 20.3% of subjects in the high-expression cohort developed a response. Moreover, an ORR of 17.8% was experienced in 45 patients who had undergone prior anti-PD-1/PD-L1 treatment. Among responders, 47% of patients were treated for >12 months, with a median treatment duration of 26.6 months (12.1–45.3) [48]. The most common AEs were asthenia (38%) and keratopathy/keratitis (38%), while grade ≥3 treatment emergent AEs occurred in 47.8% of patients and were assessed as drug related in 15.2% of the population [49].

A phase III trial evaluating the activity of CEACAM5-DM4 ADC compared to docetaxel in non-squamous-NSCLC (CEACAM5-high after the failure of standard-of-care first-line) is underway. On the other hand, in the phase II CARMEN-LC06 trial, the activity of this compound for previously treated NSCLC patients who are negative/moderate CEACAM5 expressors and with high levels of circulating-carcinoembryonic-antigen (CEA) is under investigation [50].

### 3.4. Anti-MET ADCs

C-Met is a tyrosine kinase receptor expressed on the surface of epithelial and endothelial cells. The activation of this receptor has been shown to control cell proliferation, angiogenesis, survival, and cellular motility. Dysregulation of c-Met signaling via receptor overexpression has been implicated not only in the development of NSCLC but also as a resistance mechanism [51].

Telisotuzumab-vedotin (Teliso-V) is an ADC that links the anti-c-Met humanized mAb ABT-700 with auristatin E, a potent anti-microtubule monomethyl pharmacophore [52]. The phase II Luminosity trial evaluated Teliso-V in previously treated NSCLC patients with c-Met overexpression. Overexpression was defined by immunochemistry as ≥25% 3+ (high: ≥50% 3+; intermediate: 25 to <50% 3+) in the non-squamous and as ≥75% 1+ in the squamous population. ORR was 36.5% in 52 non-squamous NSCLC EGFR wild-type patients (52.2% in the c-Met high group and 24.1% in the c-Met intermediate group) associated with a median DOR of 6.9 months, underscoring the relevance of the magnitude of expression. The most significant AEs were peripheral sensory neuropathy (25%) and nausea (22%), with grade 3 or higher AEs including pneumonia (6%), peripheral sensory neuropathy (4%), and pneumonitis (2%). Regrettably, both in the non-squamous NSCLC EGFR mutant (43 patients) and in the squamous-NSCLC cohort (27 patients), the activity was modest, with ORRs of 11.6% and 11.1%, respectively [53]. However, the association of Teliso-V with erlotinib yielded good results in a phase 1b trial in 42 patients with c-MET overexpression previously treated with EGFR-TKIs; these patients had a median PFS of 5.9 months and an ORR of 32.1% [54].

The lack of efficacy of Teliso-V in squamous-NSCLC was confirmed in the phase II Lung-MAP trial that enrolled 49 patients into 2 cohorts including ICI-naïve and ICI-refractory. This trial failed to meet the pre-specified response and pneumonitis was an observed unanticipated toxicity [55].

These findings highlight the potential of Teliso-V as a treatment option for EGFR wild-type non-squamous-NSCLC patients with MET overexpression. The phase III TeliMET NSCLC-01 trial (NCT04928846) evaluates Teliso-V compared to docetaxel in c-Met overexpressed by immunochemistry. Another ADC targeting c-Met and topoisomerase-1 overexpression, named ABBV-400, is currently being assessed in a phase I clinical trial (NCT05029882).

### 3.5. Anti-Her2 (for Non-HER2-Mutated Patients) ADCs

Human epidermal growth factor receptor 2 (HER2) is a membrane tyrosine kinase and oncogene with a well-known role in cancer due to its properties as a potent proliferative and anti-apoptotic agent. HER2 copy-number amplification was demonstrated in 2–22% of NSCLC, and HER2 overexpression was seen in 7.7–23%, with variations depending on the analytical methods used and populations examined [56].

Trastuzumab Deruxtecan (T-DXd), an ADC consisting of an anti-HER2 mAb linked to a topoisomerase-I inhibitor payload, yielded extraordinary results in metastatic breast cancer, even in HER2 low-expressors [57]. The phase II, multicenter Destiny Lung-01 trial showed T-Dxd efficacy in two cohorts of NSCLC patients refractory to standard treatment and defined by HER2 mutations or overexpression. Ninety-one patients with HER2 mutations, of whom 66% were previously treated with ICIs, obtained an ORR of 55%, a median PFS of 8.2 months, and a median OS of 17.8 months. Of note, 26% of patients experienced drug-related AEs [58]. In another study focusing on patients with HER2-overexpressing tumors, T-Dxd at 6.4 mg/kg and 5.4 mg/kg doses were evaluated. The results showed promising antitumor activity as evidenced by reported ORRs of 26.5% and 34.1% for 6.4 mg/kg and 5.4 mg/kg, respectively. Notably, the lower dose of T-Dxd demonstrated a better safety profile (drug-induced interstitial lung disease of 20.4% at 5.4 mg/kg dose versus 4.9% at 6.4 mg/kg). Consequently, further development of the compound is focused on the 5.4 mg/kg dose [59].

### 3.6. Rechallenge with Immunotherapy

#### 3.6.1. PD-1/PDL1 Inhibitor Monotherapy Rechallenge

Retreatment strategies with anti-PD-1/PD-L1 inhibitors could be effective, particularly in patients showing a good response to initial ICI treatment. Indeed, in the Keynote-010 trial, 52.4% of patients who completed 2 years of treatment with an ICI and received a second course of pembrolizumab after progression experienced clinical responses [60]. An exploratory pooled-analysis across five phase 3 trials investigating pembrolizumab recently showed a clinically meaningful benefit in this category of patients with a 6-month OS rate of 85.1%. However, it is important to note that this cohort of patients represents only a very small subgroup of the entire population. To this end, only 4.9% of patients who received pembrolizumab as a single agent and 1.8% who were exposed to a chemotherapy plus pembrolizumab combination, completed 2 years of treatment [61]. Following this exploratory pooled-analysis, Replay, an open label phase II trial, was designed to analyze the benefit of pembrolizumab rechallenge. Although preliminary data demonstrated a good safety profile with only 5.5% of grade 3 toxicity, efficacy results were modest, with a median PFS of 1.6 months and a 6-month OS rate of 59.1% [62].

In addition, in a large retrospective study, patients who had a good response to prior ICI achieved better outcomes compared to those who responded poorly: the median OS was 22.8 months for patients with a complete or partial response to prior ICI treatment versus 15.7 months for patients with stable or progressive disease [63]. It is important to point out that rechallenge does not seem a promising alternative as most patients develop resistance or do not respond to retreatment. Moreover, ICI rechallenge was associated with a decreased ORR [OR:0.29 (0.14–0.63)] and DCR [OR:0.53 (0.28–0.99)] compared with initial treatment. Therefore, careful patient selection and consideration of individual circumstances are crucial when considering retreatment with PD-1/PD-L1 inhibitors. Current data suggest that not only patients who progress after a fixed course of PD-1/PD-L1 blockers but also those who discontinued treatment due to immune-related events may benefit more from a retreatment strategy. According to a meta-analysis, both populations had numerically better ORRs and DCRs than patients who received rechallenge therapy within a maximum of 12 weeks after termination of immunotherapy (8% vs. 34% and 9% vs. 71%, respectively) [64].

#### 3.6.2. PD-1/PDL1 Inhibitors Plus Anti-Angiogenic-Agents

Emerging combinations of anti-angiogenic-agents plus ICIs have been developed based on positive results from other solid neoplasms [65].

Lenvatinib is a multitargeted-TKI of VEGF receptor 1-3, fibroblast growth factor receptor 1–4, platelet-derived-growth-factor-receptor-α, RET, and KIT. Although lenvatinib plus pembrolizumab showed promising results in a preliminary phase 1b trial with 21 NSCLC patients, the subsequent open-label phase III trial LEAP 008 comparing this combination to docetaxel in a second-line setting did not meet its primary endpoint [66,67].

Sitravatinib, a spectrum-selective-TKI targeting tumor-associated macrophage (Tyro3/Axl/MerTK) receptors and VEGFR-2, reduces the number of myeloid-derived suppressor cells and regulatory T-cells while increasing the ratio of M1/M2-polarized macrophages. It could potentially overcome an immunosuppressive TME and augment antitumor immune responses. Sitravatinib in association with nivolumab showed interesting preliminary activity in non-squamous-NSCLC patients who progressed while on chemotherapy-immunotherapy. However, a high toxic profile was reported, with the incidence of grade 3/4 adverse events reaching as high as 60% [68]. Regrettably, negative results for the phase III Sapphire trial have been posted for this combination compared to docetaxel [69].

Cabozantinib is a multi-TKI that promotes an immune-permissive environment that may enhance ICI activity. Cabozantinib was explored in the phase II Cosmic-021 trial for patients previously treated with ICIs. It had an acceptable toxicity profile and encouraging clinical activity (ORR 23%, mDOR 5.6 months, DCR 83%). However, the phase III Contact-01 trial failed to achieve a better OS compared to docetaxel [70,71].

Ramucirumab in association with pembrolizumab demonstrated efficacy in the phase I JVDF trial. Consequently, the naïve-NSCLC expansion cohort revealed an ORR of 42.3%, with higher responses in patients high PD-L1 expression [72,73]. On the basis of these promising data, the phase II Lung-MAP trial investigated this combination versus a standard of care strategy (docetaxel/ramucirumab, docetaxel, gemcitabine or pemetrexed) in 166 NSCLC patients who previously progressed on ICI treatment and chemotherapy. OS was significantly improved with the combination (median OS 14.5 months versus 11.6 months), as was the median DoR (12.9 versus 5.6 months). Indeed, OS-benefit was maintained in all subgroups (including PD-L1 and TMB) with an increased benefit in the squamous-NSCLC population (HR:0.43). On the other hand, no differences were found in terms of PFS or ORR. This novel combination exhibited a better safety profile with a grade 3 toxicity of 42% vs. 60% in the control arm. [74].

The ongoing phase III Pragmatica Lung study aims to confirm the outcomes of the Lung-MAP trial.

#### 3.6.3. PD-1/PDL1 Blockers in Association with Novel ICIs

Emerging immunotherapy agents are promising therapeutic alternatives for patients with metastatic NSCLC due to their role in acquired, innate, and humoral immunity. These drugs can act as inhibitors, stimulators, or dual modulators. Moreover, emerging ICIs can play a complementary role to current ICIs, either by involving immune cells or monitoring TME.

T-cell immunoglobulin and ITIM domain (TIGIT) is an immunomodulatory receptor that functions as an ICI in innate and adaptive immunity. It exerts direct inhibition of natural-killer (NK) cytotoxicity, T-cell activity, and competitive attenuation of CD155-mediated CD226 activation. Nevertheless, the anti-TIGIT antibody vibostolimab combined with pembrolizumab had an ORR of 3% in 38 patients with immunorefractory NSCLC [75]. Moreover, the phase 2 Keywibe-002 trial (NCT04725188), which combined pembrolizumab and vibostolimab with or without docetaxel, did not meet the PFS primary endpoint in the open-label arm [76].

Ociperlimab, another anti-TIGIT mAb, has demonstrated competent binding with C1q and Fcγ receptors inducing ADCC. Its association with tislelizumab, an anti-PD-1 mAb, produced synergistic immune cell activation and enhanced antitumor activity in preclinical models. This combination has been investigated in a phase I trial that enrolled 24 patients with previously treated metastatic solid tumors, including NSCLC patients. The combination achieved a favorable toxicity profile and responses (one partial response, nine patients with stable disease) and led to T-reg reduction, TIGIT downregulation, and proinflammatory cytokine/chemokine release [77].

T-cell immunoglobulin-domain and mucin-domain-3 (TIM-3) is a receptor expressed on myeloid cells, NK cells, and dysfunctional T-cells, and acts as an inhibitory signal often co-expressed with PD-1. The combination of sabatolimab, an antibody targeting TIM-3, and the anti-PD-1 spartalizumab, was tested in a phase Ib/II trial and showed preliminary signs of antitumor activity. Notably, 1/6 NSCLC patients enrolled developed a partial response [78]. The phase II stage of this study is ongoing for patients with NSCLC resistant to PD-1/PD-L1.

In the phase 1 AMBER study, cobolimab (a TIM-3 inhibitor) plus dostarlimab (an anti-PD-1 mAb) showed activity in terms of ORR, which ranged from 7.7% to 25% among the different dose levels, and an acceptable safety profile with 14.5% grade 3 toxicity in 55 patients with heavily pretreated tumors [79]. Consequently, the ongoing phase 2/3 COSTAR trial (NCT04655976) aims to compare the efficacy and safety of cobolimab + dostarlimab + docetaxel versus dostarlimab + docetaxel and docetaxel in an NSCLC population.

LAG-3 is a type I transmembrane protein with four Ig-like domains expressed on exhausted CD4 and CD8 tumor-infiltrating T-cells and T-regs in peripheral blood and tissue, contributing to immunoescape mechanisms. Levels of LAG-3 expression and infiltration on tumors are associated with poor prognosis [80].

Eftilagimod-alpha is a soluble LAG-3 protein that binds to a subset of MHC class-II molecules to mediate antigen-presenting cell (APC) and CD8 T-cell activation. The phase II TACTI-002 trial has studied this compound in combination with pembrolizumab. Encouraging antitumor activity was shown for naïve-NSCLC patients independently of PD-L1 expression, revealing an ORR of 37.3% and a DCR of 73.3%. Nevertheless, second-line setting outcomes for patients who had previously received anti-PD-1 were dismal, with an ORR and DCR of 8.3% and 33%, respectively [81,82].

### 3.7. Novel Immunotherapeutic Agents

#### Cytokines Blockers and Interleukins

IL-1β is normally released by various cell types, such as immune, neural, and endothelial cells. Numerous transcription factors are activated by its receptor signaling. As an example, nuclear factor κB (NF-κB) not only leads to the accumulation of tumorigenic factors in the TME but also boosts tumor-associated inflammation possibly due to the suppression of tumor growth, progression, and metastatic spread [83,84]. Unfortunately, after promising outcomes published in prior trials, the phase III Canopy-2 trial failed to prove significant efficacy for the anti-IL-1β canakinumab plus docetaxel as compared to docetaxel alone [85]. This setback highlighted the challenges associated with translating preclinical successes into clinical outcomes and underscores the complexity of targeting IL-1β signaling in the context of NSCLC treatment.

Transforming growth factor-β (TGF-β) is a cytokine that promotes tumor growth and immune evasion and progression through activity on both the innate and adaptive immune systems. TGF-β–mediated signaling in the TME acts in invasiveness, migration, and metastasis spreading via multiple mechanisms, including epithelial-mesenchymal transition. TGF-β may also impact on mechanisms of fibrosis and angiogenesis [86].

Data from a phase I expansion cohort revealed that bintrafusp-alfa had encouraging efficacy and manageable tolerability in patients with advanced platinum-experienced NSCLC not previously treated with ICIs, particularly in patients with PD-L1–high tumors. This is a first-in-class bifunctional fusion protein composed of the extracellular domain of the human TGF-β receptor II (which functions as a TGF-β “trap“) fused through a flexible linker to the C-terminus of each heavy chain of an immunoglobulin G1 antibody blocking PD-L1. Additionally, preclinical studies have shown that bintrafusp-alfa reduces TGF-β signaling within the TME.

Patients receiving the recommended dose of 1200 mg in the phase II study yielded an ORR of 37% in ICI-naïve PD-L1–positive tumors (response in 10 of 27 patients) and 85.7% (response in 6 of 7) in subjects with high-PD-L1 tumor expression. Moreover, the median PFS was 9.5 months for patients with PD-L1–positive tumors and 15.2 months for patients with PD-L1–high. Grade 3 AEs accounted for up to 29% of patients with no treatment discontinuation among the participants. Skin disorders were the most common outcome of toxicity [87].

Finally, several interleukins play an important role in immune activation. These substances enhance and expand lymphocyte activity. High doses of systemic interleukins such as IL-2 have long been explored in clinical trials. Despite promising results obtained in several types of solid tumors, the toxicity profile—including cytokine release and vascular leak syndromes—is not negligible [88]. For this reason, novel recombinant and engineered forms of some interleukins, such as IL-10, IL-15, and IL-2, have been developed and are currently being investigated in clinical trials. These modified compounds have provided interesting results in preclinical models [89].

A particularly promising drug candidate is TransCon IL-2 β/γ. This molecule is able to promote antitumor activity via NK cells and CD4 and CD8 lymphocyte stimulation while avoiding classical IL-2 side effects by not activating IL2 receptor-α, which is present in regulatory T cells, eosinophils, and endothelial cells [90]. However, the results from the TransCon IL-2β/γ phase I/II clinical trial are still pending [91].

### 3.8. Bispecific Antibodies

Bispecific antibodies (bsAbs) are antibodies that bind two distinct epitopes, and therefore combine different targets in a single antibody compound.

PD-1-CTLA4 bsAbs could become useful in the NSCLC setting because of preferential binding to CTLA-4 on PD-1-activated dual-positive T cells in the tumor (limiting toxicities in normal organs). Such antibodies initially bind to the more highly expressed receptor (PD-1) and subsequently to the second arm (CTLA-4), resulting in increased internalization and degradation of the PD-1 receptor that may result in a more durable response [92].

QL1706 is a bsAb containing a mixture of anti-PD-1 IgG4 and anti-CTLA-4 IgG1. It was first tested in a phase I trial on 519 patients with advanced solid tumors. Among the 149 patients with NSCLC, ORR, and mDoR were 14%, with a better ORR achieved in immuno-naive patients (24%). QL1706 was well tolerated, with TRAEs and immune-related AEs of grade ≥3 occurring in 16.0% and 8.1% of patients, respectively [93]. SI-B003 is another PD-1/CTLA4 bsAb that was evaluated in a phase I trial in advanced solid tumors; this trial revealed a DCR of 52% among 21 patients who had received prior PD-1/PD-L1 treatment [94].

PD-1-TIGIT bsAbs. The bsAb AZD2936 is a humanized IgG1 targeting PD-1 and TIGIT. This novel compound demonstrated preliminary activity in Artemide-01, an open-label phase I/II trial enrolling patients with PD-L1-positive advanced-NSCLC. All included patients had previously experienced disease progression on anti-PD-1/PD-L1 therapy. In preliminary data from the first 76 evaluable patients, three showed a partial response and thirty had stable disease. Treatment-related AEs were recorded in three patients only, indicative of a good safety profile [95].

EGFR-HER3 bsAbs. EGFR and HER3 are both highly expressed in various epithelial tumors, including NSCLC [96]. The EGFR-HER3 bsAbs, SI-B001, was evaluated in a phase II open-label trial for 55 patients with advanced NSCLC EGFR/ALK wild-type. Among 22 patients evaluable in cohort B, SI-B001 plus docetaxel exhibited an ORR of 45.5% and DCR of 68.2%, with a manageable toxicity profile as a second-line therapy after previous chemotherapy-IO [97].

BL-B01D1 is another novel ADC consisting of an EGFR-HER3 bsAb linked to a TOP-I-inhibitor payload via a cleavable linker. A phase I trial investigated its role in many cancer types. BL-B01D1 demonstrated encouraging efficacy, not only in EGFR-mutated NSCLC, but also in 42 NSCLC EGFR wild-type patients who progressed while on chemotherapy-IO (ORR 40.5%, DCR 95.5%) [98].

### 3.9. Cellular Immunotherapy

The presence of tumor infiltrating lymphocytes (TILs) in the tumor and its peritumoral compartment has been proposed as an immunotherapy biomarker, with numerous studies showing a positive correlation between TILs and good prognosis. Interestingly, not only the presence of TILs but also their differentiation and localization have been shown to determine clinical outcomes in diverse tumors [99,100]. TIL therapy has already exhibited efficacy and feasibility in a multicentric phase III study for melanoma patients [101].

A single-arm open-label phase I trial was conducted to evaluate the efficacy of TILs administered with nivolumab in 20 previously treated NSCLC patients. Autologous-TILs were expanded ex-vivo from minced tumor tissue and cultured with IL-2. Patients received cyclophosphamide and fludarabine lymphodepletion, followed by TIL infusion and IL-2. Maintenance treatment with nivolumab was subsequently given. Promising results were described for 13 evaluable patients, with 11 patients having a reduction in tumor burden (median best change of 35%). Furthermore, three of those patients developed a complete response, with two of them in remission 1.5 years later [102] (Table 2 and Table 3).

Chimeric antigen receptor (CAR)-T cell immunotherapy also provides a new approach for the treatment of NSCLC, even if results do not seem satisfactory like hematological malignancies because of immunosuppressive TME. Many phase I/II trials are underway for NSCLC including several surface antigens (EGFR, CEA, HER2 and many others) [103]. TCR-engineered T-cell therapy also seems promising. Nowadays, most of these T-cells are engineered to recognize only one antigen and most clinical trials in NSCLC are targeting cancer-germline antigens, showing good tolerability but modest efficacy [104].

### 3.10. Vaccines

Active immunotherapy with vaccines has been developed in the past few years. Tedopi is a neoepitope vaccine restricted to HLA-A2-positive patients who represent approximately 45% of the Caucasian NSCLC population. This vaccine targets five tumor-associated antigens commonly expressed in NSCLC: ACE, HER2, MAGE2, MAGE3, and P53.

In the phase III trial Atlante-1, Tedopi was compared to chemotherapy in pretreated NSCLC patients. Tedopi was superior in terms of HR (0.59 [0.38–0.91]) and median OS (11.1 versus 7.5 months). Although Tedopi did not demonstrate improvements in PFS or ORR, this vaccine showed a gain in post-progression survival and time-to-worsening performance status in patients who had primary or secondary resistance to immunotherapy as the last treatment [105]. The phase II Combi-TED trial is now underway to evaluate Tedopi combined with either docetaxel or nivolumab versus docetaxel alone for NSCLC patients progressing on first-line chemotherapy-immunotherapy [106].

## 4. Future Perspectives

Despite the unprecedented survival benefit of PD-1 axis blockade in advanced-NSCLC, most patients fail to achieve durable responses, even in tumors with high-PD-L1 expression. Indeed, the landscape of possible treatments after failure on first-line chemotherapy-immunotherapy remains a complex challenge.

Docetaxel is presently considered the standard of care, albeit with modest gains in OS, which may potentially be marginally enhanced by a post-immunotherapy effect based on retrospective analyses.

According to current data, retreatment with anti-PD-1/PD-L1 blockers is unlikely to be a useful strategy for most NSCLC individuals who previously progressed while on these compounds. Nevertheless, the approach may be a viable option for a highly select subgroup of patients. Individuals with an ICI-free interval of at least 12 months or those who previously discontinued anti-PD-1/PD-L1 blocker treatment due to immune-related events might be the most suitable candidates for immunotherapy resumption (Figure 1).

In the post-immunotherapy context, anti-angiogenic agents associated with chemotherapy could be of some interest, potentially enhancing the effect of previous immunotherapy. This combination holds special interest for younger patients or those who experienced a shorter PFS on first line treatment. One plausible approach might be weekly-paclitaxel chemotherapy plus bevacizumab in non-squamous-NSCLC patients that assures better PFS and ORR than docetaxel, with modest toxicity effects as well. Nevertheless, the combination of anti-angiogenic agents with ICIs does not seem to be as efficient as expected given the negative outcomes of phase III trials.

ADCs are certainly the most promising drugs, not only for NSCLC patients but also for many other individuals with distinct solid tumors. Owing to rapid improvements in bioengineering and conjugation technologies, these novel compounds exhibit increased potency and a broader range of targetable tumors. Further studies are still necessary to improve conjugation features, optimize physicochemical properties, manage toxicity profiles, identify surface proteins for antibody binding, and explore combinations with ICIs. Based on available data, it is our opinion that ADCs will soon replace chemotherapy as second-line standard of care for NSCLC.

On the other hand, we suggest that novel immunotherapeutic agents or new combinations of ICIs with other compounds might be an appealing therapeutic strategy to treat NSCLC patients. It is crucial to emphasize that selective biomarker identification beyond PD-L1 status is still needed to enhance personalized treatment approaches and overcome specific immunoresistance.

## 5. Conclusions

In conclusion, we consider that personalized combination strategies that are developed according to the pathways or hallmarks that specifically drive each patient’s tumor biology will remain the main challenge. Consequently, we believe that reconsidering tumor mutational status with tissue re-biopsy or liquid biopsy after immunotherapy failure could represent an intriguing approach. Indeed, particularly in cases where next generation sequencing has not been completed, this approach could lead to the identification of previously undiscovered actionable alterations whose therapy is generally addressed with second-line options. Alternatively, potential mechanisms of resistance may be identified which could increase the chances of patients receiving better treatment options or being enrolled in clinical trials assessing specific treatment algorithms in well-defined patient populations.

## Figures and Tables

**Figure 1 cancers-15-05505-f001:**
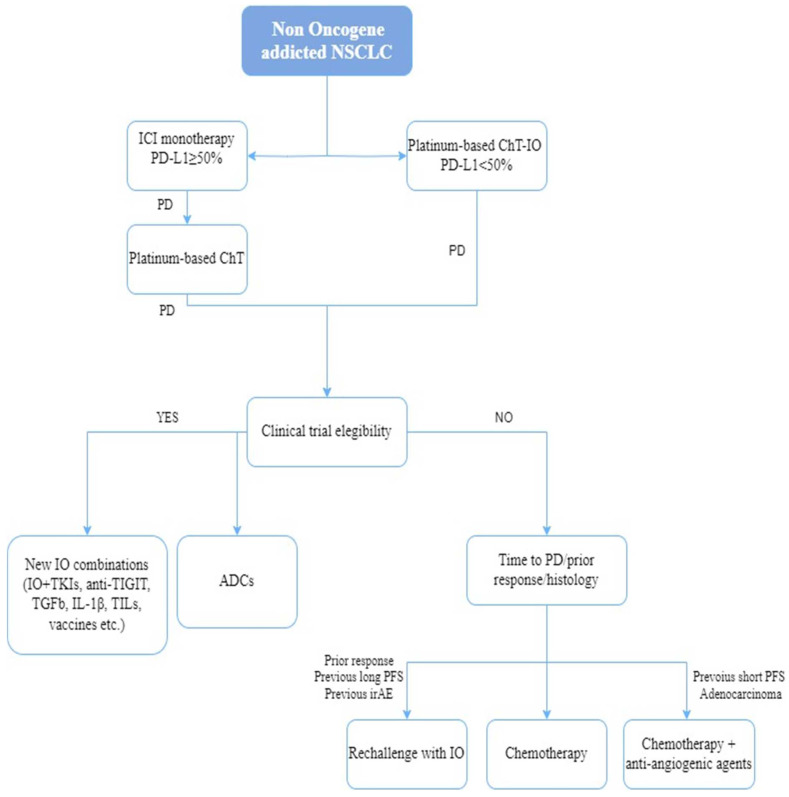
Therapeutic algorithm for second-line metastatic non-oncogene addicted NSCLC. AA = anti angiogenic agents; ADCs = antibody drugs conjugates; ChT = chemotherapy, IO = immunotherapy; ChT = chemotherapy, irAE = immune-related adverse event; mono= monotherapy; NSCLC = non-small cell lung cancer; PD = progression disease; PFS = progression free survival.

**Table 1 cancers-15-05505-t001:** Current options after progression on first-line therapy, including ICIs.

Study (Ref.)	Design	Treatment	Population	Nr.	Outcomes
Shepherd FA. et al. [13]	Prospective Randomized	Docetaxel vs. BSC	Progressed on first-line platinum-based chemotherapy	104	PFS 10.6 vs. 6.7 weeks OS 7.0 vs. 4.6 months
Schuette W. et al. [14]	Phase III	Docetaxel w. vs. 3 weekly	Progressed on first-line platinum-based chemotherapy	215	OS 6.3 vs. 9.2 months ORR 10.5% vs. 12.6%
Crinò L. et al. [15]	Phase II	Gemcitabine	stage IIIB or IV	83	mDoR 29 weeks
Yoneshima Y et al. [17]	Non inferiority Phase III	Nab-paclitaxel vs. Docetaxel	Progressed on first-line platinum-based chemotherapy	503	OS 16.2 vs. 13.6 months PFS 4.2 vs. 3.4 months ORR 29.9% vs. 15.4%
Hanna N, et al. [18]	Non inferiority Phase III	Pemetrexed vs Docetaxel	Progressed on first-line chemotherapy	571	PFS 2.9 vs 2.9 months OS 8.3 vs 7.9 months
Nokihara H, et al. [19]	Non inferiority Phase III	S-1 vs. Docetaxel	Progressed on at least first-line platinum-based chemotherapy	1154	PFS 2.86 vs. 2.89 months OS 12.75 vs. 12.52 months
LUME-Lung 1 Reck M. et al. [28]	Phase III	Docetaxel ± Nintedanib	Progressed on first-line platinum-based chemotherapy Adenocarcinomas	655	PFS 3.4 vs. 2.7 months OS 10.1 vs. 9.1 months OS 12.6 vs. 10.3 months
VARGADO Grohè C. et al. [29]	Prospective no interventional	Nintedanib + Docetaxel	Previously treated with chemotherapy-IO (cohort C)	135	DCR 72.5% PFS 4.8 months
REVEL Garon EB. [30]	Phase III	Ramucirumab ± Docetaxel	Progressed on first-line platinum-based chemotherapy	1825	PFS 4.5 vs. 3 months OS 10.5 vs. 9.1 months
ULTIMATE Cortot AB. et al. [32]	Phase III	Paclitaxel weekly vs. Docetaxel	Previously treated with 1 or 2 prior lines	166	PFS 5.4 vs. 3.9 months ORR 22.5% vs. 5.5% OS 9.9 vs. 10.8

DCR = Disease Control Rate; ORR = Overall Response Rate; OS = Overall Survival; PFS = Progression Free Survival; mDoR = median duration of response.

**Table 2 cancers-15-05505-t002:** Emerging options in metastatic NSCLC previously treated with ICIs.

Ph.	Treatment	Population	Nr.	Outcomes
I/II	Datopotamab Deruxtecan	Previously treated NSCLC unselected for TROP-2	180	ORR 26% mDOR 10.5 months mPFS 6.9 months mOS 10.4 months
III	Datopotamab Deruxtecan vs Docetaxel	Pretreated patients with advanced/metastatic NSCLC	604	ORR 26.4% vs. 12.8% PFS 4.4 vs. 3.7 months mDoR 7.1 vs. 5.6 months OS 12.4 vs. 11 months
I/II	Sacituzumab Govitecan	Previously treated NSCLC unselected for TROP-2	54	ORR 19% mDoR 6 months mPFS 5.2 months mOS 9.5 months
II	Tusamitamab Ravtansine	Non-squamous previously treated moderate CEACAM5 high CEACAM5	28 64	ORR 7.1% ORR 20.3%
II	Telisotuzumab Vedotin	Metastatic NSCLC, ≤2 prior lines, ≤1 line of chemotherapy, MET amplification	136	No-squamous: ORR 36.5% mDoR 6.9 months Squamous: ORR 11.1% mDoR 4.4 months
II	Trastuzumab Deruxtecan 5.4 mg/kg	HER2 amplified NSCLC	41	ORR 34.1% mPFS 6.7 months mOS 11.7 months
	Trastuzumab Deruxtecan 6.4 mg/kg		49	ORR 26.5% mPFS 5.7 months mOS 12.4 months
Ib/II	Lenvatinib + Pembrolizumab	Previously treated NSCLC	21	ORR 33% mDoR 10.9 months mPFS 5.9 months
II	Sitravatinib + Nivolumab (no-squamous)	Non-squamous NSCLC previously treated with chemotherapy and ICIs	68	ORR 16% mPFS 6 months mOS 15 months mDoR 13 months
II	Cabozantinib + Atezolizumab	Progression to prior ICI and ≤2 prior lines of systemic therapy excluding VEGFR TKI	81	ORR 19% mDoR 5.8 months DCR 80% mPFS 4.5 months mOS 13.8 months
III	Cabozantinib + Atezolizumab vs. Docetaxel	Previously treated with chemotherapy and ICIs	366	mOS 10.7 vs. 10.5 months mPFS 4.6 vs. 4.0 months ORR 11% vs. 13.3%
II	Ramucirumab + Pembrolizumab vs Investigator choice chemotherapy	Previously treated with chemotherapy and ICIs	166	mOS 14.5 vs. 11.6 months mPFS 4.5 vs. 5.2 months ORR 22% vs. 28% mDoR 12.9 vs. 5.6 months
I	Vibostolimab + Pembrolizumab	Previously treated NSCLC	38	ORR 3%
I	Ociperlimab + Tisletizumab	Pan tumor, previously treated	24	1 PR 9 SD
I/Ib	Sabatolimab + Spartalizumab	Previously treated NSCLC	6	1/6 PR
II	Eftilagimod alpha + Pembrolizumab	PD-1/PDL1-resistant NSCLC	36	ORR 8.3% DCR 33%
I	TILs + Nivolumab	Previous immunotherapy	20	11 pts reduction tumor burden 3 confirmed responses
III	Canakimumab + Docetaxel vs. Docetaxel	Previous chemotherapy + immunotherapy sequential or concomitant	237	mOS 10.5 vs. 11.3 months mPFS 4.17 vs. 4.21 months
I	Bintrafusp alfa	Previous platinum-based chemotherapy PD-L1 positive PD-L1 high	80	ORR 25% ORR 36% ORR 85.7%
III	Tedopi vs. Docetaxel or Pemetrexed	HLA2 + Previous chemotherapy + immunotherapy sequential or concomitant	118	mOS 1.1 vs. 7.5 months ORR 8% vs. 18% mPFS 2.7 vs. 3.4 months 6m DCR 25% vs. 24%
I	QL1706	Previously treated NSCLC Immuno naïve	149	ORR 14% mDoR NR ORR 24%
I	BL-B01D1	Previously treated NSCLC	42	ORR 40.5%, DCR 95.5%

DCR = Disease Control Rate; mDoR = median Duration of Response; mOS = median Overall Survival; mPFS = median Progression Free Survival; ORR = Overall Response Rate; PR = partial response; pts = patients; SD = stable disease.

**Table 3 cancers-15-05505-t003:** Ongoing phase II and III trials testing novel compounds.

Study	Ph	Treatment	Population	Nr.
NCT05555732	III	Datopotamab Deruxtecan + Pembrolizumab with or without chemotherapy	No Prior Therapy for Advanced or Metastatic PD-L1 TPS < 50% nonsquamous NSCLC without Actionable Genomic Alterations	975
NCT05215340	III	Datopotamab Deruxtecan + Pembrolizumab vs. Pembrolizumab	Treatment-naïve Subjects with Advanced or Metastatic PD-L1 High (TPS ≥ 50%) NSCLC without Actionable Genomic Alterations	740
NCT05089734	III	Sacituzumab Govitecan vs. Docetaxel	Metastatic NSCLC with Progression on or After Platinum-Based Chemotherapy and Anti-PD-1/PD-L1 Immunotherapy	580
NCT05186974	II	Sacituzumab Govitecan + Pembrolizumab with or without platinum-based chemotherapy	First-line Treatment of Patients with Metastatic NSCLC Without Actionable Genomic Alterations	224
NCT04154956	III	SAR408701 vs. Docetaxel	Previously Treated, CEACAM5 Positive Metastatic nonsquamous NSCLC	450
NCT05245071	II	SAR408701	Nonsquamous NSCLC Participants with Negative or Moderate CEACAM5 Expression Tumors and High Circulating CEA	38
NCT04524689	II	SAR408701 + Pembrolizumab with or without platinum-based chemotherapy	CEACAM5 Positive Expression Advanced/Metastatic nonsquamous NSCLC not previously treated	120
NCT04928846	III	Telisotuzumab-Vedotin vs. Docetaxel	Previously Treated c-Met Overexpressing, EGFR Wildtype, Locally Advanced/Metastatic nonsquamous NSCLC	698
NCT05513703	II	Telisotuzumab-Vedotin	Previously Untreated MET Amplified Locally Advanced/Metastatic nonsquamous NSCLC	70
NCT03976375	III	Lenvatinib + Pembrolizumab vs. Docetaxel	Previously Treated Metastatic NSCLC and Progressive Disease after Platinum Doublet Chemotherapy and Immunotherapy	405
NCT05633602	III	Ramucirumab + Pembrolizumab vs. standard of care	Previously Treated with Immunotherapy for Stage IV or Recurrent NSCLC	700

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
