# Peer review of "Future Perspectives in the Second Line Therapeutic Setting for Non-Oncogene Addicted Non-Small-Cell Lung Cancer"

_cancers, 2023, doi:10.3390/cancers15235505_

Round 1

Reviewer 1 Report

Comments and Suggestions for Authors

This review summarizes the current and future state of therapy in the second-line treatment of driver gene mutation- and translocation-negative NSCLC. It is well organized and comprehensive.

1. The efficacy of chemotherapy after ICI is discussed mainly by ORR, but the OS of DTX+RAM does not seem to differ significantly between ICI-treated and untreated patients (Garon EB, et al. Front Oncol 2023; Sep 4: 13: 1247879. PMID 37731641). There also is a retrospective study reporting similar results (Kato R, et al. J Immunother Cancer 2020 Feb; 8(1): e000350. PMID 32066647). It would be desirable to discuss not only on ORR but also on PFS and OS.

2. A phase III trial comparing nab-paclitaxel with docetaxel has been reported (Yoneshima Y, et al. J Thorac Oncol. 2021 Sep;16(9):1523-1532. PMID: 33915251) and seems to warrant consideration for inclusion in the list.

3. As for cellular therapy, clinical trials of CAR-T are underway for various types of cancer, including NSCLC, and TCR-engineered T-cell therapy is also under development. It may be worth mentioning because CAR-T and TCR-modified T-cell therapy are expected to become more mainstream than TIL infusion therapy in the future.

4. Line 71: Keynote is usually all capitalized (KEYNOTE).

5. Line 224: The first letter of anti- should be capitalized along with the others.

Author Response

We are thankful to the reviewer for the feedback around the organization e contents of our manuscript.

  1. The efficacy of chemotherapy after ICI is discussed mainly by ORR, but the OS of DTX+RAM does not seem to differ significantly between ICI-treated and untreated patients (Garon EB, et al. Front Oncol 2023; Sep 4: 13: 1247879. PMID 37731641). There also is a retrospective study reporting similar results (Kato R, et al. J Immunother Cancer 2020 Feb; 8(1): e000350. PMID 32066647). It would be desirable to discuss not only on ORR but also on PFS and OS.

Thank you for this suggestion. We reported data of phase II/III trials in this setting but we really believe that these suggestions will help us to clarify the current poor efficacy of chemotherapy with or without anti-angiogenics after ICIs treatment.

This section now reads:

A systematic review of 10 studies suggests a numerical better PFS of docetaxel plus ramucirumab in prior exposed patients to ICI than in those naïve to these compounds (5.7 vs 3.8 months) with statistical significance in two trials (P=0.012 and P=0.041). However, according to this publication, docetaxel plus ramucirumab could not offer OS advantage in ICI pretreated patients .Hence, another retrospective observational study, including 1439 patients, reported that patients treated with chemotherapy with or without anti-angiogenics after ICIs had better ORR [HR 1.71 (1.19-2.46); p = 0.004] without any OS [HR 1.05 (0.86-1.28), p=0.63] and PFS advantage [HR 0.95 (0.8-1.12), p 0.55], compared to patients ICI-untreated.

  1. A phase III trial comparing nab-paclitaxel with docetaxel has been reported (Yoneshima Y, et al. J Thorac Oncol. 2021 Sep;16(9):1523-1532. PMID: 33915251) and seems to warrant consideration for inclusion in the list.

That’s a good point. The introduction of this non-inferiority phase III trial in our list will allow us to make our review more complete. We have attached the new version of Table 1 in a Word file.

3. As for cellular therapy, clinical trials of CAR-T are underway for various types of cancer, including NSCLC, and TCR-engineered T-cell therapy is also under development. It may be worth mentioning because CAR-T and TCR-modified T-cell therapy are expected to become more mainstream than TIL infusion therapy in the future.

               We totally agree. CAR-T and TCR-modified T-cell therapy are two future options in NSCLC. We mentioned as follow:

Chimeric antigen receptor (CAR)-T cell immunotherapy also provides a new approach for the treatment of NSCLC, even if results does not seem satisfactory like hematological malignancies because of immunosuppressive TME. Many phase I/II trials are underway for NSCLC including several surface antigens (EGFR, CEA, HER2 and many others).TCR-engineered T-cell therapy also seems promising. Nowadays, most of these T- cells are engineered to recognize only one antigen and most clinical trials in NSCLC are targeting cancer-germline antigens, showing good tolerability but modest efficacy.

  1. Line 71: Keynote is usually all capitalized (KEYNOTE).
  2. Line 224: The first letter of anti- should be capitalized along with the others

The point 4 and 5 in line 71 and 224 have been modified as suggested.

Reviewer 2 Report

Comments and Suggestions for Authors

The submitted manuscript is an engaging review that provides a comprehensive overview of the current therapeutic landscape, focusing on second-line treatments for non-oncogene-addicted non-small cell lung cancer (NSCLC) after the progression of initial therapies. The introduction offers a solid background, setting the stage for the subsequent discussions.

The paper extensively covers second-line drugs for NSCLC, delving into potential future scenarios such as antibody-drug conjugates, combination immunotherapy with anti-angiogenic agents or novel immune checkpoint inhibitors, vaccines, bispecific antibodies, cytokine blockers, and interleukins. The molecular mechanisms of action are elucidated, accompanied by an introduction to major clinical trials. Notably, the inclusion of studies with negative outcomes, like CONTACT-01 and SAPPHIRE in the immunotherapy with anti-angiogenic agents section, and CANOPY-2 in the cytokine inhibitors section, adds completeness to the discussion.

Regarding a minor issue, there is a suggestion to update the data on Dato-DXD. In the manuscript, reference is made to a press release, but in reality, the Tropion-Lung01 data were presented at the ESMO 2023 plenary. These data demonstrated a significant advantage in progression-free survival (PFS) over Docetaxel (4.4 months versus 3.7 months; hazard ratio [HR] 0.75; 95% confidence interval [CI] 0.620.91; p=0.004), with a trend of advantage in overall survival (OS) (12.4 months with Dato-DXD and 11.0 months for docetaxel, HR 0.90; 95% CI 0.721.13). Additionally, confirmed objective response rates (ORR) were 26.4% with Dato-DXD and 12.8% with docetaxel, with median durations of response of 7.1 months and 5.6 months, respectively, in previously treated patients with advanced NSCLC (LBA12).

In conclusion, the paper emphasizes that developing personalized combination strategies based on the specific pathways driving each patient's tumor biology remains a significant challenge. The suggestion of reconsidering tumor mutational status with tissue re-biopsy or liquid biopsy after immunotherapy failure is an intriguing approach. Considering the minor issue mentioned above and the overall importance of the submitted manuscript, the paper could be accepted.

Author Response

 We are thankful to the reviewer for the feedback about our manuscript. As you can imagine we prepared this paper before the ESMO 2023 presentation. As a consequence, we added the results of this trial in table 2 (attached in a Word file) and included in the text as follow:

The phase III Tropion LUNG-01 trial, which compared Dato-DXd to docetaxel in patients previously treated with ICIs, showed positive results with an ORR of 26.4% vs 12.8%, a PFS of 4.4 vs 3.7 months [HR: 0.75 (0.62–0.91); p=0.004] with a trend of advantage in OS.

We are thankful for the opportunity to amend and improve our work and we now resubmit a revised copy of the manuscript that contains all the major amendments highlighted during the process of peer review.

We have done our utmost to address all the reviewers’ comments but would be happy to accommodate any further changes as necessary.

We feel that our manuscript is significantly strengthened after addressing these comments, and we hope that our revised manuscript is now acceptable for publication.
